# Concussion Office Based Rehabilitation Assessment: A Novel Clinical Tool for Concussion Assessment and Management

**DOI:** 10.3390/brainsci10090593

**Published:** 2020-08-27

**Authors:** Matthew Katz, Stephane Lenoski, Haitham Ali, Neil Craton

**Affiliations:** Department of Family Medicine, Sport and Exercise Medicine, University of Manitoba, Winnipeg, MB R3T 2N2, Canada; umlenoss@myumanitoba.ca (S.L.); dr.haithamali@hotmail.com (H.A.); dcraton@mpi.mb.ca (N.C.)

**Keywords:** concussion, sport related concussion (SRC)

## Abstract

The Concussion Office Based Rehabilitation Assessment (COBRA) is a novel tool constructed to ensure a comprehensive assessment of patients who may have sustained a concussion. The SCAT-5 (Sport Concussion Assessment Tool) has long been the gold standard for concussion assessment, however, it was designed as a sideline tool and its utility can be seen to decrease after a few days post-concussion. It also does not prompt evaluation of all the phenotypes of concussion. As such, the COBRA was created to assess the majority of potential manifestations of concussion in the office setting a day or two after an injury has been sustained. The COBRA utilizes the eight phenotypes of concussion as a guide to assess each of the potential biopsychosocial features that can be associated with these injuries and can be used to guide evidence-based treatments. Through early identification of concussion phenotypes, the clinician may start optimal treatment and hopefully prevent prolonged recovery and persisting symptoms.

## 1. Introduction

Concussion is an important public health problem in the domain of sport and exercise medicine. Sport related concussion (SRC) is common, involving approximately 3.8 million athletes per year in the United States. It is noted that most patients have symptom resolution in the first 7–10 days after injury [1], although this have come into question with several recent publications [2]. SRC is defined by the Berlin Consensus Statement as “immediate and transient symptoms of traumatic brain injury induced by biomechanical forces”. It is characterized by a rapid onset of short-lived neurologic impairments that will resolve spontaneously. These symptoms are generally not associated with structural changes, however they may result in neuropathological impairments. SRC may or may not involve a loss of consciousness. In most cases, sequential improvement is the natural history [3]. The American Society of Sport Medicine defines concussion similarly as a “traumatically induced transient disturbance of brain function that involves a complex pathophysiologic process” [4]. Similar definitions are seen with the Veterans’ Affairs, as well as American College of Rehabilitation Medicine definitions [5,6]. There are more concussive injuries outside of the domain of athletics, than those attributed to athletics alone [7]. Falling is the most common etiology [7]. Concussion is a clinical syndrome involving multiple symptoms. Leslie et al. proposed that this syndrome does not require brain involvement, as symptoms may be the result of non-neurologic pathology such as whiplash or inner ear pathology [8]. This influences the clinical assessment and treatment of patients with potential concussive injury.

Craton et al. attempted to classify concussion based on the seven distinguishing phenotypes [9]. Similar efforts to subdivide the clinical manifestations of concussion have been proposed by other researchers [10,11]. From the work done by these groups, it can be purported that concussion manifests in at least seven different ways: cognitive dysfunction, oculomotor symptoms, affective disturbances, cervical dysfunction, headache, cardiovascular manifestations and vestibular manifestations (COACH CV). Furthermore, recent research has shown the consistent relationship between somatic symptom disorder and prolonged concussive symptoms [12]. A clinical tool to accurately assess SRC in the health care providers office and determine the appropriate phenotype would be an asset for all clinicians, especially those not treating patients with these injuries on a regular basis.

The current gold standard for concussion evaluation is the Sideline Concussion Assessment Tool 5 (SCAT-5). This is a standardized tool designed for assessment of potential SRC in the milieu of the sporting event by those covering the sporting event. It is designed for the immediate or on-field assessment after an athlete sustains an injury. The SCAT-5 prompts reviewing red flags, observable signs, memory, Glasgow coma scale, as well as assessment of the cervical spine [13]. The purpose of the SCAT-5 is that of a screening tool to merit a more detailed clinical assessment. It is not meant as a diagnostic assessment, nor is it appropriate outside the realm of sport. The utility of the SCAT-5 assessment is seen to decrease as soon as three days after injury [14]. At this time, there is no comprehensive, validated tool that can be used in the health care provider’s office to guide the clinical assessment a day or two after a patient has sustained a potential concussion. Given the multiple potential biomedical and psychosocial issues that can be involved in the evolution after head and neck injury, a focused clinical tool to evaluate these areas is warranted.

The goal of this paper is to introduce a novel clinical inventory that purports to evaluate the specific potential phenotypes of concussion. The Concussion Office Based Rehabilitation Assessment (COBRA) has been designed to assist the clinician in the identification of potential impairments and treatment opportunities that have presented themselves in the first few days after injury.

The COBRA (see Figure 1) is designed to serve as a roadmap for the clinician to assess, document and treat each specific phenotype of concussion. It uses the acronym COACHCV, from Craton et al., with the addition of further classification for somatic symptoms, using the Somatic Symptom Scale 8. It provides a comprehensive method of assessing the patient in the office, not in the milieu of sport. Of note, the COBRA is not a substitute for a comprehensive neurologic examination. It is designed as a tool to allow for further assessment of concussion phenotypes. In the author’s experience, the observable characteristics of concussed patients tend to cluster around one or two manifestations, such as where headache is the dominant symptom, or dizziness is the main complaint [15]. The COBRA itemizes information that is important in the assessment of injured individuals when concussion is suspected. The clinical time required to consider each of these potential trajectories is justified by the current literature [15].

The COBRA is divided into sections in order to characterize each phenotype of concussion based on symptoms and clinical signs of that area. The first section is a symptom inventory designed to evaluate the burden of potential concussion symptoms using the familiar Likert scale. The maximum symptom score is 132. A higher symptoms score is associated with a worse prognosis, so documenting this information is important for the clinician [1]. Within this section, four questions overlap with somatic symptom disorder as tested with Somatic Symptom Scale-8 and they have been denoted separately. Similar assessment has been done with the Rivermead test; however, the benefit of the COBRA is in the further subcategorization of concussion phenotypes. Rivermead would be a suitable screening test for cognitive concussive symptoms.

Cognitive impairment in the form of memory concerns, decreased attention and concentration are some of the most common symptoms of concussion. This is seen to be especially prominent in the acute period [1]. This is associated with decreased mental processing speed as well as impaired executive function. The SCAT-5 can be utilized to assess cognitive function in the acute period. However, as previously explained, the utility decreases after the first 3–5 days. Other tests that have attempted to measure cognitive performance include ImPACT and Cogsport testing [16]. These tests are not available in most clinical scenarios. As a result, there is a need for a clinical assessment tool to help the clinician accurately diagnose concussion. The COBRA allows the clinician to test cognitive function through orientation, memory, recall and attention. Deficits in the cognitive assessment can trigger the clinician to focus on cognitive rehabilitation. This may involve the use of neuropsychological interventions if symptoms persist [1].

Oculomotor manifestations have frequently been associated with concussion [6,13]. Dysfunction in near point convergence, saccadic eye movements and visual pursuit have all been described. In some studies, over 60 per cent of patients with concussion reported symptom provocation with oculomotor screening [6,13,17]. Furthermore, when oculomotor dysfunction has been recognized, it has been shown to be a prognostic indicator associated with prolonged recovery [6,18]. The COBRA prompts the testing of near point convergence, saccadic eye movements, visual pursuit and extra-ocular movements. Deficits in these domains can focus the clinician on ocular rehabilitation. The purpose of this test is simply to establish if there are normal or abnormal extraocular movements.

Affective disturbances are a risk factor for persisting concussion symptoms. They can also be a potential consequence of concussion and ten of the traditional concussion symptoms are also symptoms of depression (feeling slowed down, difficulty concentrating, fatigue or low energy, trouble falling asleep, more emotional, irritability, sadness, nervousness or anxiety) [9]. As such, the COBRA suggests emphasis of this trajectory of concussion. Using answers from the symptom scoring index, as well as specific questions addressing psychiatric symptoms, the clinician is able to establish if an affective trajectory of concussion is present. Once this is seen, careful patient follow-up is indicated. Depression and anxiety are commonly seen as co-morbidities in the concussed individual and can be consequences after concussion [9]. Early treatment is essential.

There is significant overlap between the clinical presentation of concussion and that of whiplash [8]. It is important to recognize that concussion patients may present with a cervicogenic source to many of their symptoms. The prevalence of cervical pathology in concussion is unknown, however, cervical spine dysfunction has been documented as one of the most prominent phenotypes of concussion. Matuszak, concluded that the examination of the cervical spine, neck range of motion, Spurling test and palpation of the muscular and bony anatomy was important in assessing concussion patients [19]. The COBRA utilizes these tests, as well as incorporating the Canadian C-Spine rules in order to assess for more worrisome cervical spine injuries. With focused rehabilitation of the cervical spine and its musculature, the patient can be treated effectively and restored to normal function earlier [19].

Headache is perhaps the most common symptom associated with concussion. Most types of headaches can be seen in the concussion patient. Red flags must always be reviewed with concussion for significant intra-cranial pathology, however, once this has been ruled out, treatment of headache can be based on the specific type of headache documented. The most common headaches associated with concussion have been migraine, tension, post-traumatic and cervicogenic [20,21]. Pre-existing migraines have been associated with poorer outcomes post-concussion. Some headaches may benefit from pharmacologic management, as such, early recognition is important in treatment. Physiotherapy for cervicogenic headache has been shown to expedite recovery [22].

One of the lesser known trajectories of concussion is cardiovascular dysfunction. There are several known associated cardiovascular manifestations of concussion. These include exercise intolerance, altered heart rate variability, postural orthostatic tachycardia syndrome (POTS), autonomic nervous system anomalies, elevated heart rate and others [23,24,25]. Therefore, it is important to assess the patient for these signs. The COBRA suggests questions for exercise intolerance and the documentation of orthostatic symptoms related to heart rate and blood pressure which could be reflective of autonomic dysfunction. By establishing this trajectory early, rehabilitation can be focused on improving cardiovascular function. This may be done with the use of submaximal heart rate training as well as gradual aerobic exercise training [11].

Vestibular dysfunction is a common manifestation of concussion. Up to 81% of patients shows vestibular dysfunction on initial presentation with concussion [26]. The vestibular system receives multiple inputs from multiple systems including the oculomotor, brainstem, spinal cord, cervical spine, cerebral cortex, cerebellar and peripheral sensory systems. Patients with concussion may present with vertigo, dizziness, balance and gait difficulties as well as associated visual symptoms. Through testing with Rhomberg, tandem gait, Dix-Hallpike, BESS (Balance Error Scoring System) testing, visual motion sensitivity and the presence of nystagmus, the clinician is able to assess possible vestibular dysfunction. When established, this patient may be suited for vestibular rehabilitation [8,10,20].

The final potential phenotype observed with concussion is somatization. Assessing patients in the acute setting for Somatic Symptom Disorder (SSD) seems less intuitive, but pre-injury somatic symptom score has been shown to be a strong predictor of symptom duration in concussed patients [27]. The concussion Likert scale shares several of the same symptoms with SSD and it may be difficult to distinguish between the two, especially in protracted cases. While of less value in the first few weeks after injury, SSD can be associated with delayed recovery. Symptoms related to this disorder can be attributed to brain injury and some studies have shown up to 55% of patients with delayed recovery are somatising [28]. Patients with Somatic Symptom Disorder often require specific psychological intervention. The COBRA utilizes the Somatic Symptom scale-8. Four symptoms of potential somatization are found in the general concussion symptom inventory, and the other four symptoms are included in its own section. The total score of symptoms is added to be out of 48, with a higher score indicated increased somatization behavior.

## 2. Discussion

While the SCAT-5 is designed for the sideline assessment of concussion, the majority of concussion patients are diagnosed in the clinical setting. Given that the SCAT-5 sensitivity decreases 2–3 days after the incident, and the majority of clinic visits do not occur in this period, there is a need for a different method of assessment for this milieu [1]. The COBRA encompasses a detailed roadmap for assessing concussion and its various phenotypes.

The COBRA has been developed to identify most manifestations of concussion and all potential phenotypes. Treatments now exist for all phenotypes of concussion so identifying specific clinical impairments can help patients move past the “rest until symptoms abate” treatment nihilism of previous guidelines [9,18,29].

The COBRA offers a more comprehensive office based assessment than that stipulated by the SCAT-5. Questions in the SCAT-5, such as the Maddock’s questions that evaluate the athlete’s recall of events in a sporting contest, do not translate well to the office-based setting and are not relevant to non-sport related concussion. Clinical manifestations in the domains of the oculomotor system, affective disturbances, vestibular signs, cardiovascular anomalies, subtypes of headache and somatization are not prompted by the SCAT-5. All these difficulties have been documented in concussed patients, can affect prognosis, and be meritorious of treatment [6,9,10,11,21,23,24,25,26,29].

Further research on the validity and reliability of the COBRA for diagnosis of concussion and management is required. It needs to be field tested by a wide range of health care providers. While, it has been evaluated by a small group of physicians and athletic therapists and the value seen, further study will be needed to determine if this tool is a useful instrument. The COBRA is a simple instrument which allows clinicians from a variety of disciplines to evaluate a patient with a potential concussion in a systematic and organized fashion. It provides a helpful road map to the clinician not well versed in concussion assessment to ensure a comprehensive evaluation is undertaken. It provides an excellent record of the clinical examination that can be easily incorporated into an electronic medical record.

## 3. Conclusions

The COBRA introduces a novel inventory for the clinical assessment of concussion. Its value is in determining if concussion has occurred and the specific phenotype involved. With early diagnosis, the patient can be treated specifically with the goal of early return to function. Further work is required to demonstrate the sensitivity, specificity and diagnostic utility of the COBRA.

## Figures and Tables

**Figure 1 brainsci-10-00593-f001:**
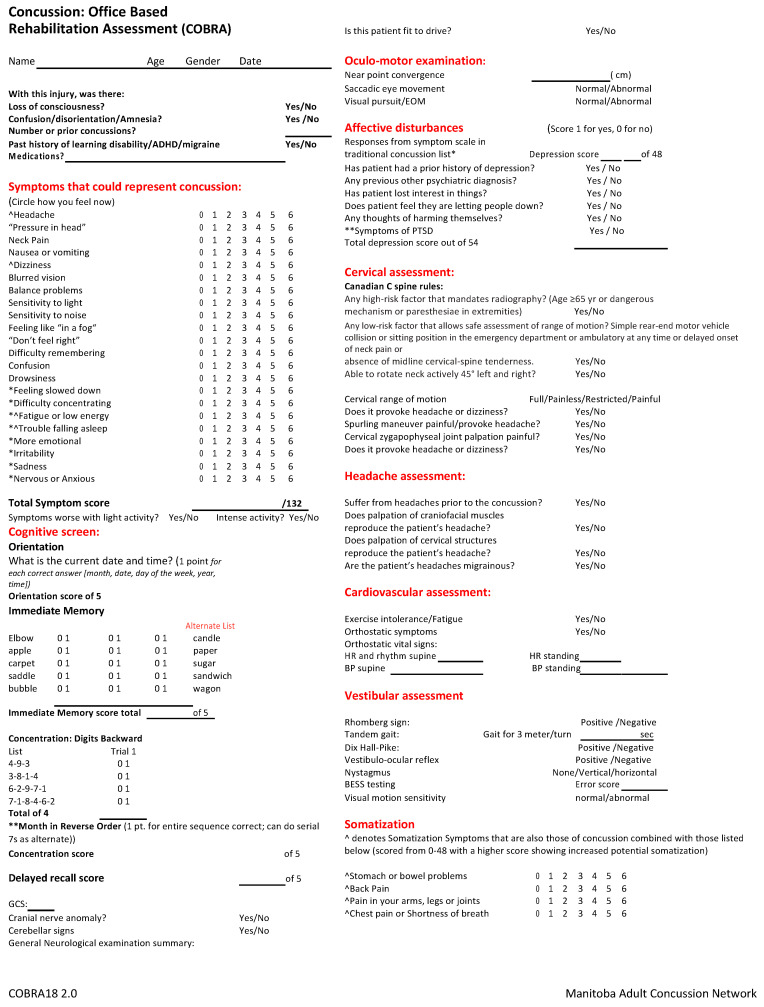
The Concussion Office Based Rehabilitation Assessment (COBRA).

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
