# Peer review of "Concussion Office Based Rehabilitation Assessment: A Novel Clinical Tool for Concussion Assessment and Management"

_brainsci, 2020, doi:10.3390/brainsci10090593_

Round 1

Reviewer 1 Report

Introduction, Line 35 - the authors need to clarify whether they mean there are more concussive injuries outside of athletics than those attributed to athletics alone, or if they mean that non-athletic concussions are another understudied mechanisms in which there are additional incidences of concussion, with appropriate references. 

It also remains unclear why/how this form is valid 2-3 days post injury and the SCAT-5 is not.  The body of the manuscript includes a discussion of each part of the COBRA, however, the discussion section is lacking. Perhaps consider further discussion of how the COBRA compliments existing clinical assessments and why the instrument may enhance clinical assessment of concussion outside of sport-specific mechanisms. 

Author Response

Thank you for the opportunity to improve our manuscript and your attention to detail.   We have clarified the comment on the epidemiology of concussion with an appropriate reference. We have added some commentary on why the SCAT 5 loses validity after a few days.  It is noteworthy, that the concussion in sport group provides this quotation without citation. We have added some discussion on how we feel the COBRA can be a useful to to complement current assessment practice.   Thanks again.

Reviewer 2 Report

I have no new comments

Author Response

No response necessary

Round 2

Reviewer 1 Report

Thank you for providing tracked changes. A few minor comments below: 

Line 37: This line needs a reference.

Line 40: Clarify what is meant by 'this group'. The sentence reads as if little is known about patients who sustain a concussion from falls. 

Lines 62-64: This added sentence seems to belong before the preceding sentence as its not related to the decreased utility of the SCAT-5 but rather its applicability outside of sports. 

Lines 80-82: Is there any literature to support the author's experience? If so, please reference. 

Lines 84-85: This sentence needs references. 

Lines 122-123: This sentence needs a reference.

Line 199-203: Please change "all" to "a variety". The last sentence (lines 202-203) is a subjective opinion statement and should be removed. 

Author Response

Thank you very much for your comments.  We have edited the lines and added the appropriate references.  Thanks for the feedback.

This manuscript is a resubmission of an earlier submission. The following is a list of the peer review reports and author responses from that submission.

Round 1

Reviewer 1 Report

The manuscript is a concept paper really interesting, especially for the clinicians involved in concussion, for whom is devoted the scale. Obviously, the absences of numerical data modifies the usual way of peer-review. However, there are some minor questions that the authors must change to improve the manuscript.

1.- The number of the first cite is the 20 and the second the 27. All the numbers are misplaced. Please, indicate the cites in a correct order. 

2.- Please, clarify what is the meaning for functional at line 29. Are they meaning that these functional symptoms aren’t organic?. Usually, functional is synonymous to psychogenic.

3.- Line 36. Add the number of cite for Leslie et al.

4- Line 74. Explain in a little more detail what potential trajectories are. How do they define trajectories, by dynamic evolution in time or by the groups of symptoms?.

5.- Lines 165-166. Is there any publication (e,g congress proceedings) with the results indicated at this sentence?. It would be helpful to provide it.

Reviewer 2 Report

Abstract, Line 12 – Has this been demonstrated in the literature? If so, reword to, “and its utility has been shown to decrease…”

Abstract, Line 14 – All is a strong word here.  Can you sure that you have an exhaustive list of possible manifestations?  Perhaps it is more appropriate to say, “for a more comprehensive assessment of potential manifestations…” or something along those lines.

Introduction, Line 22 – I advise introducing with the public health impact of concussion rather than the popularity of the topic.

Introduction, Lines 41-44 – It is unclear where the acronym is coming from.

Introduction, Lines 63-65 – This sentence needs to be reframed as an objective statement.

I feel the actual form is better suited as an appendix.

There appears to be a heading missing before the description of the content of the COBRA.

The description of the COBRA includes several discussion statements that are currently presented as opinion and not supported by literature, as presented. For example, Lines 92-94 could be restated with appropriate references to justify why a cognitive assessments would inform use of cognitive rehabilitation and/or neuropsychological interventions. Please revise throughout. 

More discussion on how the COBRA form improves upon the SCAT-5 needs to be added, including a discussion of its application to non-sporting populations and utility in the clinic 2-3+ days post-injury. The qualitative discussion of the physician testing does not provide objective data on the usability of the form, nor the number of physicians who have tested the form.  This section is not overly helpful or informative. 

Correct formatting of references according to the journal guidelines – some are inside punctuation, some are outside

There are several grammatical errors throughout the document that need to be addressed.
